# Understanding the Mechanisms Driving Fibrosis Following Cochlear Implantation—Lessons from Other Tissues

**DOI:** 10.3390/cells14231924

**Published:** 2025-12-03

**Authors:** Cecilia M. Prêle, Kady J. Braack, Marcus Atlas, Jafri Kuthubutheen, Tylah Miles, Wilhelmina H. A. M. Mulders, Steven E. Mutsaers

**Affiliations:** 1School of Medical, Molecular and Forensic Sciences, Murdoch University, Murdoch, WA 6150, Australia; 2Institute for Respiratory Health, The University of Western Australia, Nedlands, Perth, WA 6009, Australia; 22511924@student.uwa.edu.au (K.J.B.); admin@resphealth.uwa.edu.au (T.M.); steven.mutsaers@uwa.edu.au (S.E.M.); 3School of Human Sciences, The University of Western Australia, Crawley, Perth, WA 6009, Australia; helmy.mulders@uwa.edu.au; 4Ear Science Institute Australia, Nedlands, Perth, WA 6009, Australia; marcus.atlas@uwa.edu.au; 5Centre for Ear Sciences, Medical School, The University of Western Australia, Nedlands, Perth, WA 6009, Australia; 6Division of Surgery, Medical School, The University of Western Australia, Nedlands, Perth, WA 6009, Australia; jafri.kuthubutheen@uwa.edu.au; 7Department of Otolaryngology Head and Neck Surgery, Sir Charles Gairdner Hospital, Perth, WA 6009, Australia; 8Pharmacology and Toxicology, School of Biomedical Sciences, Nedlands, Perth, WA 6009, Australia

**Keywords:** cochlear implant, fibrosis, anti-fibrotic, scarring, foreign body response, neo-ossification

## Abstract

Cochlear implants are highly successful in restoring speech perception but variability in outcomes exists. Post-surgical fibrosis and neo-ossification are thought to play a significant role, being linked to increased impedance and loss of residual hearing and posing challenges for re-implantation. Hence, there is growing interest in pharmacological interventions to limit intracochlear fibrosis and neo-ossification. While current approaches focus on steroids, studies in other organs have identified many candidate drugs. However, selection is hindered by a limited understanding of the molecular and cellular mechanisms driving fibrosis after implantation. This review introduces potential drug candidates for cochlear implant-induced fibrosis, with many targeting core fibrotic pathways such as TGF-β/SMAD, PDGF, and Wnt/β-catenin or inhibiting pro-inflammatory signalling. By drawing on lessons from other tissues, this review identifies mechanisms and therapeutic approaches adaptable to the cochlea. Understanding fibrosis across organs will guide strategies to prevent or reverse cochlear fibrosis. Their translation requires careful evaluation of local delivery, minimal ototoxicity, and effects on the electrode–tissue interface.

## 1. Introduction

Hearing restoration surgery has advanced considerably since the first cochlear implant in 1961 [1], but variability in speech perception outcomes has been reported, with the underlying causes not yet fully understood [2]. Current cochlear implant candidacy includes both individuals with profound deafness and those with residual natural hearing [3]. Maintenance of residual hearing and hearing quality in these individuals has been a major research focus which has led to significant changes in cochlear implant electrode design, composition, and surgical approach. A key factor thought to influence cochlear implant performance is the formation of fibrotic scar tissue and neo-ossification around the implanted electrode. Histopathological analyses of post-mortem temporal bones from cochlear implant recipients consistently reveal varying degrees of fibrosis and ossification surrounding the electrode [4,5,6,7,8,9,10]. Although the effect of implant-induced fibrosis on hearing outcomes remains unclear, these studies indicate that most, if not all, cochlear implant patients develop some degree of intracochlear fibrosis.

Tissue damage from physical trauma, infection, metabolic insults, or ageing disrupts cellular homeostasis and, depending on the type, location and duration of injury, can markedly impact tissue function. Furthermore, the capacity for tissue repair depends on the intrinsic capacity of the cells within that tissue to proliferate or self-renew. Tissues and cells without this capacity heal through a process of scar formation. The cellular and molecular events driving wound healing are tightly regulated to ensure resolution of inflammation, effective repair and to minimise the formation of fibrotic scars. Pathological fibrosis is thought to occur as a consequence of repetitive injury that drives the chronicity of the response through the perpetuation of an exaggerated, overexuberant inflammatory and deregulated wound healing response, which ultimately results in excessive and disorganised extracellular matrix (ECM) deposition at sites of injury [11].

Fibrosis and scarring occur in many clinical settings and are a significant and growing clinical problem. In a global burden of disease analysis, liver cirrhosis and other chronic liver diseases, interstitial lung disease (ILD) and pulmonary sarcoidosis, chronic kidney disease and ischaemic heart disease, were reported to be responsible for 46.2, 3.77, 41.5 and 182 million DALYs (disability adjusted life years), respectively, worldwide [12]. Furthermore, it is estimated that the annual combined global incidence of fibrotic diseases, including heart, liver, kidney, lung, skin and systemic fibrotic disorders is ~4968/100,000 people years [13]. Although the pathogenesis of cochlear fibrosis is not well understood, detailed mechanistic studies on scarring and fibrosis in other organs, along with transcriptome and proteome analyses during fibrosis initiation, have revealed common pathways driving fibrogenesis across tissues. This raises the potential for repurposing existing therapies to treat cochlear fibrosis.

## 2. Fibrosis in the Inner Ear

In the inner ear, fibrotic scarring is known to occur as a consequence of infections, such as bacterial meningitis [14,15]. Infection is a leading cause of moderate to severe acquired sensorineural hearing loss, which is often permanent. For example, bacterial meningitis can lead to haring loss in up to 32% of children and 69% of adults [16]. Hearing loss may result from the spread of bacterial products and inflammatory mediators through the meninges and cerebrospinal fluid via the cochlear aqueduct [17] or as a consequence of the development of fibrosis. For example, cytomegalovirus infection-induced hearing loss in a mouse model is linked to the development of fibrosis and aberrant macrophage function in the cochlea [18,19]. Furthermore, a clinical study performed using the National Temporal Bone, Hearing and Balance Pathology Resource Registry Database, identified new bone formation or fibrosis in all cases of autoimmune associated hearing loss and in 20–30% of cases of labyrinthitis or meningitis [20]. Sensorineural hearing loss was also identified in 40% of patients with systemic sclerosis (SSc), an autoimmune-mediated fibrotic connective tissue disease with multi-organ involvement [21]. Data from a histological case study also suggested that both the middle and inner ear can be affected by SSc, with evidence of reduced numbers of outer hair cells, dendrites and ganglion cells in the cochlea and perivascular fibrosis linked to vasculitis [22]. One case study also suggested that auditory neuropathy, the functional loss of synapses and/or auditory nerve fibres, occurred as a result of SSc [23]. However, a recent systematic review on the relationship between SSc and hearing loss concluded that although most papers suggest a higher incidence of hearing impairment in patients with SSc, there is a lack of sufficient evidence to provide a firm estimate of prevalence and insight in underlying mechanisms [24].

Significant intracochlear fibrosis and ossification also occurs following cochlear implant surgery, with inflammation driving the formation of dense fibrotic tissue around the electrode array, a process driven by foreign body and wound healing responses initiated by trauma and damage to inner ear tissue at the time of surgery [25] (summarised in Figure 1). A recent clinical study reported cochlear fibrosis/ossification in the basal turn of the cochlea in 5 of 17 (29.4%) cochlear implant revision surgeries [26]. Post-mortem temporal bone studies place this number much higher with fibrosis and ossification observed in almost all individuals who have undergone cochlear implant surgery in their lifetime [4,5,25,27,28]. A recent 3-dimensional imaging study performed on post-mortem temporal bones from 20 patients showed evidence of fibrosis and ossification in all cochleae, and on average, 33% of the cochlear volume was taken up with fibrous and osseous tissue [6].

The specific impact that fibrosis or ossification has on the function of the cochlear implant remains largely unknown. Interestingly, in individuals with autoimmune or immune-mediated sensorineural hearing loss, fibrosis can occur because of systemic disease, and this has been linked to higher variability in cochlear implant outcomes in this population [29]. Several studies have suggested that fibrosis and neo-ossification after cochlear implantation have a detrimental effect on hearing measures in both patient and animal studies. Indeed, animal studies demonstrate a negative relationship between the extent of new tissue growth after cochlear implantation and hearing measures, including auditory brainstem response thresholds and spiral ganglion cell counts [30,31,32]. Several human studies corroborate these data, showing a correlation between fibrosis and neo-ossification with a worsening of hearing measures, including word recognition scores and auditory thresholds [27,33,34]. The potential relationship between fibrosis and hearing loss after cochlear implantation is particularly significant in patients with partial hearing loss receiving hybrid cochlear implants. Within this patient demographic, all studies show an immediate deterioration of residual hearing post-surgery and further hearing loss at various degrees and rates in the long-term [35,36,37,38]. This poses a significant challenge, as electric-acoustic stimulation in cochlear implant users with residual hearing offers substantial benefits over electrical stimulation alone, including improvements in speech [39] and music perception [40,41]. While residual hearing loss may have several causes, including excitotoxicity in sensory cells after inflammation and mechanical trauma to the microstructures of the cochlea, fibrosis also seems to be a significant factor. A case study of post-mortem temporal bone of a cochlear implant user who lost residual hearing after implantation found that fibrosis within the cochlea was the most likely explanation for this residual hearing loss [33]. This is likely due to the effect of fibrosis on cochlear mechanics. Indeed, using mathematical modelling, Choi et al. demonstrated that efforts to preserve residual hearing clinically may be limited by the changes in mechanotransduction caused by intracochlear fibrosis, which dampens the wave response of the basilar membrane [42]. Several animal model studies in different species have confirmed implant-induced tissue reaction in the cochlea [43,44,45,46,47]. O’Leary and colleagues previously demonstrated in an animal model that progressive hearing loss after cochlear implant surgery is linked to the extent of the implant-induced tissue reaction [32] and that low frequency hearing loss can be delayed by altering the surgical approach to implantation [48].

Given evidence that fibrosis in the inner ear increases variability in hearing outcomes following cochlear implantation and may adversely affect residual hearing, a longstanding challenge has been to identify surgical and pharmacological strategies to minimise fibrosis and preserve residual hearing. However, clinical studies have been hampered by the lack of audiological markers for fibrosis. Retrospective analysis of post-mortem temporal bone tissue and linked clinical data suggest electrode impedance and word recognition score as surrogate measures of fibrosis. For example, Kamakura et al. showed a negative correlation between consonant-nucleus-consonant word score and the percentage of intracochlear new bone formation, and this was related to the amount of trauma to the basilar membrane at the time of surgery [27]. A recent study found that higher four-point impedances, reflecting the local environment at the neural-tissue interface, along the electrode array correlate with poorer hearing outcomes. The author suggested that these changes represent physiological alterations in basilar membrane mechanics, likely resulting from fibrotic changes within the cochlea [49]. This aligns with data from a guinea pig model, which demonstrated a correlation between increased impedance and the extent of fibrosis and neo-ossification [50]. While higher impedances have been linked to increased fibrosis, clinical quantitative data correlating electrode impedance and the extent of fibrosis and/or hearing loss remain limited. Heutink et al. performed one of the first studies to correlate the extent of implant-induced neo-ossification with residual hearing. Using ultra-high-resolution CT, they showed that new bone formation led to long-term loss of residual hearing in the low-frequency range [51]. Overall, these studies further support the negative impact intracochlear tissue reaction has on hearing and highlights the need to limit the extent of fibrosis and bone formation post implant.

## 3. Neo-Ossification

The relationship between neo-ossification and fibrosis after cochlear implantation remains unclear. Neo-ossification is classified as metaplastic, arising from surgical disruption of the inner ear blood supply, or osteoplastic, involving osteoblasts and developing after endosteal damage [52]. Post-implantation neo-ossification is likely osteoplastic. Although it occurs in several inner ear pathologies, including meningitis, otosclerosis, otitis media and trauma, it is unknown whether its development after implantation is directly or indirectly linked to fibrosis [53]. Geerardyn et al. [6] observed a correlation between implant duration and cochlear neo-ossification, but not with fibrosis, suggesting it may represent progressive ECM deposition with fibrosis transitioning to bone [4]. Although some mechanisms driving fibrosis and neo-ossification after cochlear implantation are known, further research is required to clarify these pathways and support development of targeted strategies to reduce both complications.

Like fibrosis, post-implantation neo-ossification may reflect dysregulated wound healing, particularly when the cochlear lateral wall is damaged [28,54]. Mesothelial cells can also transdifferentiate into osteoblastic cells [55], potentially contributing to both fibrosis and neo-ossification. In a mouse model, Bas et al. reported increased expression of bone-associated genes, including bone morphogenic protein 4 (BMP4), connective tissue growth factor (CTGF), and fibroblast growth factor (FGF)7 [56], with higher CTGF levels where fibrosis bordered neo-ossification or bone [57].

Although some mechanisms driving fibrosis and neo-ossification after cochlear implantation are known, further research is required to clarify these pathways and support development of targeted strategies to reduce both complications.

## 4. Fibrosis in Other Tissues

In most tissues, fibrosis is part of normal wound healing and, in skin, helps maintain barrier integrity. However, excessive dermal scarring can impair thermoregulation, limit tissue movement, cause pain and cosmetic distress, and lead to psychological effects [58]. While most scars remodel, some individuals develop hypertrophic scars with excessive, disordered collagens, or keloids that grow beyond the wound boundary [59].

In organs such as the heart and lung, excessive ECM deposition severely impairs function. For example, after myocardial infarction, ischaemic reperfusion injury leads to functional cardiomyocytes being replaced with scar tissue, disrupting electrical conduction [60]. In the lung, disorganised collagen within the interstitium restricts mechanical function and gas exchange [61]. Treatments to reduce fibrosis are limited. Some interventional approaches, such as pulse field ablation, may help reduce post-infarct arrhythmias [62,63], while antifibrotic drugs like pirfenidone (PFD), have been designed to target transforming growth factor-beta (TGF-β)-driven fibroblast activation [64,65,66,67,68]. In idiopathic pulmonary fibrosis (IPF), both pirfenidone (PFD) and nintedanib (NTD) target fibrosis progression by inhibiting key profibrotic signalling pathways [69,70], although their clinical utility is often constrained by adverse side effects [71,72,73]. Chronic kidney disease is a major global issue, with tubulointerstitial fibrosis the key process leading to functional decline and kidney failure [74]. Abnormal repair involving endothelial, epithelial, mesangial inflammatory and immune cells drives TGF-β-mediated ECM deposition, capillary rarefaction and hypoxia, promoting fibrosis [74,75]. Drugs blocking the renin-angiotensin system, endothelin-1 and vasopressin signalling can slow progression, but no effective antifibrotic therapy exists despite many mechanistic targets under trial [76].

Liver fibrosis results from chronic liver diseases including non-alcoholic fatty liver disease, alcoholism and viral hepatitis [77]. Hepatocyte-derived danger-associated molecular patterns (DAMPs) activate Kupffer and other immune cells, driving hepatic stellate cell transdifferentiation into collagen-producing myofibroblasts [78]. Excess ECM disrupts liver architecture, and as with other fibrotic diseases, no drugs can reverse with transplantation the best option [79].

Fibrosis can also extend beyond the originating tissue, as in keloids or post-operative adhesions, where aberrant serosal healing creates fibrous bands between organs and the body wall. Adhesions restrict abdominal and intrapleural movement, cause pain, and account for most intestinal obstructions and many infertility cases [80,81]. They are dynamic structures containing fibroblasts, blood vessels, smooth muscle and nerves [80], likely formed via cellular invasion and mesothelial cell trans-differentiation [55,82,83]. Fibrosis also forms around implanted medical devices, such as fibrotic sheaths around pacemaker leads [84] due to the foreign body response, which encapsulates foreign material to protect the host [85].

## 5. Mechanisms of Fibrosis

While the cellular mechanisms driving fibrosis have been thoroughly characterised in some disease settings, others, such as cochlear implant-induced fibrosis, remain comparatively understudied, with only limited investigations to date.

A defining feature of fibrosis is the accumulation of fibroblasts and myofibroblasts at sites of tissue damage or repair. These cells secrete large amounts of ECM proteins, particularly collagen. Myofibroblasts in particular are highly synthetic and play a central role in ECM production and wound contraction [86]. These cells are normally removed by apoptosis during the resolution phase of normal wound healing, however in fibrosis they persist [87,88,89]. In lung fibrosis, myofibroblasts are thought to acquire a senescent phenotype, characterised by reduced mitogenic activity but increased resistance to apoptosis [89,90,91]. The origin of myofibroblasts remains incompletely understood, but several potential sources have been proposed. These include the proliferation, recruitment, and differentiation of resident fibroblasts, as well as contributions from pericytes, endothelial cells, circulating fibrocytes drawn to the injury site, and epithelial cells undergoing epithelial-to-mesenchymal transition (EMT) [92]. EMT is driven by high levels of pro-fibrotic growth factors, particularly TGF-β1, produced at the injury site by resident and immune cells such as macrophages. There is also evidence that monocytes and macrophages can contribute to the myofibroblast pool by differentiating through a macrophage-to-myofibroblast transition (MMT) [93]. Similarly, mesothelial-to-mesenchymal transition (MesoMT) may serve as a source of myofibroblasts in serosal fibrosis, including conditions such as encapsulating peritoneal sclerosis and post-operative adhesion formation [83,94]. Within the liver, hepatic stellate cells are the predominant collagen secreting cells activated by immune cells and pro-fibrogenic mediators to differentiate into myofibroblasts [95]. Fibroblasts, myofibroblasts and macrophages all play a synergistic roll in the propagation of an abnormal tissue repair response, which can progress to fibrosis. In cochlear implantation, a driving factor for dysregulated wound healing is the implant itself, wherein the continued presence of a foreign body leads to chronic activation of the immune system, fibrosis and neo-ossification. In the inner ear, the spiral ligament contains abundant fibrocytes that play an important role in ECM deposition and fibrosis [96,97]. In addition, studies using models of Keyhole Limpet Hemocyanin-induced immune labyrinthitis and platinum wire cochlear implantation in rat showed that cochleae became filled with fibroblastic tissue derived from the mesothelial cells lining the scala tympani and vestibuli [98]. These findings suggest that mesothelial cells may contribute to local fibrosis following cochlear implantation.

Macrophage accumulation is thought to drive the early fibrotic response through secretion of profibrotic mediators which lead to the accumulation, differentiation and activation of fibroblasts/myofibroblasts within the scala [99,100], possibly through MesoMT [83]. Indeed, pro-fibrotic mediators including TGF-β, CTGF, FGF and fibronectin, have been detected as early as 3 h and up to 30 days after cochlear injury or implant insertion [56,101,102,103,104]. Notably, a recent study by Rahman and colleagues reported that macrophage depletion did not alter the extent of fibrosis following cochlear implantation in mice [105]. This suggests that while macrophages are key to the early inflammatory and immune response after implantation, they may not be the primary drivers of fibrosis. Instead, as in other tissues, the accumulation of fibroblastic cells at the cochlear implant site appears to be mediated by pro-fibrotic factors such as TGF-β, CTGF, platelet-derived growth factor (PDGF), and FGF [57,101].

## 6. Current Prevention and Treatments for Cochlear Implant-Mediated Fibrosis

Fibrosis and/or neo-ossification occurs in nearly all cases of cochlear implantation in both humans and animal models, typically arising at the surgical site and around the implant [5,106,107]. Increasing evidence that fibrosis impairs device function and residual hearing in hybrid (electric-acoustic) cochlear implants highlights the urgent need for strategies to prevent or reduce implantation-induced fibrosis.

### 6.1. Surgical

Early strategies to limit cochlear implant-associated fibrosis focused on minimising mechanical trauma to the cochlea and its microstructures during surgery and implant insertion [108]. Advances aimed at reducing surgical trauma include the use of round window (RW) insertion instead of cochleostomy, as well as softer, more flexible implant designs [109,110]. These ‘soft surgery’ techniques have been associated with reduced new tissue growth and improved residual hearing [109,111]. Despite the advancements in surgical techniques, fibrosis still limits the therapeutic benefits of implantation in some patients, suggesting that a multi-modal approach, combining surgical refinements with pharmacological interventions, may be necessary to more effectively prevent fibrosis.

### 6.2. Corticosteroids

Pharmacological modulation of cochlear fibrosis has largely centred around the use of steroids. Corticosteroids are anti-inflammatory synthetic analogues of steroid hormones produced in the adrenal glands [112]. Because postoperative inflammation is associated with fibrosis, neo-ossification, and elevated hearing thresholds, corticosteroids have been evaluated for their potential to limit these adverse tissue responses [113]. Corticosteroid treatment in humans and animals with cochlear implants, whether topical or systemic, short or long term, has been shown to preserve residual hearing and reduce fibrosis [114,115,116,117]. A recent systematic review of human studies reported a tendency for perioperative steroids to have a positive impact, at least in the short term, on hearing and vestibular function preservation in individuals following cochlear implantation [118]. Recent studies have also shown improved preservation of spiral ganglion neurons (SGN) after local dexamethasone elution in chronically stimulated animals [119]. A study using dexamethasone-eluting cochlear implant electrodes in guinea pigs demonstrated significant reductions in fibrotic tissue and electrode impedance compared to untreated controls [117]. Similarly, in humans, cochlear perfusion with the steroid triamcinolone resulted in significantly lower long-term electrode impedance compared to controls [120], indicating reduced fibrosis. In contrast, the systemic delivery of methylprednisolone did not reduce electrode impedance spikes [121], suggesting the anti-inflammatory action of steroids is most effective when locally administered. A more recent study using dexamethasone eluting arrays in a cochlear implant model in guinea pigs, showed inhibition of neo-ossification and reduced SGN loss; however, no effect was observed on the extent of fibrosis [122]. Similarly, a study in non-human primates demonstrated beneficial effects of a dexamethasone-eluting electrode array on tissue reaction and impedance levels [123]. In contrast, studies in a guinea pig model showed no change in monopolar electrode impedance over five weeks with dexamethasone-eluting arrays, and no significant differences in fibrotic tissue formation, new bone growth, or SGN density between array types [50]. However, four-point impedance was significantly lower in the presence of dexamethasone, and this reduction correlated with the percentage of fibrous tissue and new bone growth [50]. Furthermore, reduced electrode impedance in patients which received a dexamethasone eluting electrode as compared to conventional electrodes has been observed in two separate small cohort clinical trials [124,125]. Thus, there is currently a lack of consensus on the beneficial effect of steroids on cochlear implants; however, steroid eluting electrodes hold some promise by providing direct and sustained intracochlear drug delivery but the need for effective drug targeting approaches remains.

## 7. Novel Therapeutic Approaches for Cochlear Fibrosis Informed by Insights from Other Tissues

A growing body of research has delineated the cellular processes and signalling pathways driving tissue fibrosis, including inflammatory pathways, TGF-β/Mothers against decapentaplegic homologue (Smad), tyrosine kinase (TK) receptors, lysyl oxidase, interleukin (IL)-6/signal transducer and activator of transcription (STAT)3, and Wnt/β-catenin signalling, among others [126,127,128]. These pathways have since become targets for therapy, and many of the anti-fibrotic drugs currently approved for treatment target some of these pathways (summarised in Figure 2). This review will discuss four pathways that are being targeted for anti-fibrotic therapies in various tissues, with potential application in cochlear fibrosis: inflammation, TGF-β/Smad, Wnt/β-catenin and TKs.

### 7.1. Inflammation

The role of inflammation and the immune system in fibrosis across various tissues has been widely debated. However, growing evidence now supports that the inflammatory and immune responses play crucial roles in both the development and progression of the fibrotic process [129,130,131]. Many inflammatory mediators and receptors have been identified as being actively involved in fibrosis. For example, the receptor for advance glycation end products (RAGE), is considered to be an important mediator, responsible for driving the inflammatory response in fibrosis. RAGE is a membrane receptor and a member of the receptor immunoglobulin superfamily. There are many variants of RAGE and notably soluble RAGE (sRAGE) is a potential serum biomarker in fibrosis [132]. RAGE activation plays an important role in EMT and DNA repair in lung fibrosis. RAGE expression is decreased in IPF patients [133] and RAGE-knock-out mice develop pulmonary fibrosis [134]. However, the effects of RAGE appear to be organ specific as opposite effects have been described in the liver. For example, RAGE activation in hepatocytes has been shown to stimulate TGF-β and monocyte chemoattractant protein 1 (MCP-1) production and induce fibroblast alpha smooth actin (αSMA) expression and collagen production, and blocking RAGE pathway activation reduced hepatitis C-induced fibrosis [135]. Similarly, SiRNA-mediated RAGE inhibition led to reduced production of profibrotic mediators IL-6, tumour necrosis factor (TNF) and TGF-β [136] and suppressed the development of hepatic fibrosis [137].

### 7.2. TGF-β/Smad Pathway

One of the key mediators of fibrosis is TGF-β, which is produced by various resident and immune cells, including airway epithelial cells (AECs) and macrophages, and is also present in the latent form within the ECM. TGF-β expression is upregulated in nearly all fibrotic tissues, and for many years, therapeutic approaches have focussed on targeting TGF-β-driven responses [138]. TGF-β is upregulated following cochlear implantation both in human cochlear fibrotic tissue [57] and in the cochleae of implanted mice [56]. As a central mediator of fibrosis, TGF-β exerts its pro-fibrotic effects through either the canonical Smad-dependent or non-canonical Smad-independent pathways and exacerbates fibrogenesis by inducing differentiation of cells into myofibroblasts and inducing ECM production and deposition, in particular collagen [139,140]. The Smad family comprises eight transcriptional regulators downstream of TGF-β receptors, with Smad2, Smad3 and Smad4 acting as the principal effectors. Of these, only phosphorylated Smad3 can directly bind Smad-binding elements within gene promoters to drive transcription, whereas phosphorylated Smad2 and Smad4 lack DNA-binding domains and instead modulate Smad3-dependent activation of fibrogenic genes such as type I collagen [138,141]. In contrast, Smad7 antagonises TGF-β/Smad3 signalling by competing with Smad3 for DNA-binding, making its induction a potential anti-fibrotic strategy [138,142]. TGF-β also activates several non-canonical pathways, including p38 mitogen-activated protein kinase (MAPK) and phosphatidylinositol-3-kinase (PI3K)/Akt, both implicated in EMT [143]. Additionally, Rho-like GTPases promotes cytoskeletal remodelling and mesenchymal traits [144], effects that may be amplified by TGF-β-induced reactive oxygen species (ROS) generated through nicotinamide adenine dinucleotide phosphate (NADPH) oxidase 4 [145]. ROS can further enhance TGF-β signalling by augmenting Smad activity independently of inflammasome activation [146].

The Janus kinases (JAKs) are receptor-associated TKs that play a role in cytokine and growth factor signalling. Activation of JAK1 by TGF-β leads to early pro-fibrotic tyrosine-phosphorylation of STAT3, independent of Smads [147,148]. However, late phase activation of STAT3 crucially requires integration with Smad3 for proliferation, myofibroblast differentiation, and enhancing TGF-β-mediated transcription of key pro-fibrotic genes [148].

Various approaches have been trialled to block TGF-β signalling and ablate fibrosis with limited effect. The most conventional approach is to block TGF-β and TGF-β receptor interaction using receptor antagonists or monoclonal antibodies. These strategies have shown success in animal models of fibrosis, including those affecting the liver, lung, kidney and heart [141,149,150,151], but have produced limited benefits in clinical trials [152,153]. Likewise, inhibition of Smad3 or overexpression of Smad7 reduces fibrosis in animal models of liver and lung disease and in systemic conditions such as SSc [154,155,156], yet this approach remains difficult to translate clinically.

The most successful anti-fibrotic drug thought to work by inhibiting TGF-β is PFD. PFD is a non-peptide synthetic molecule with well-established anti-fibrotic and anti-inflammatory properties. Although its precise mechanism of action remains unclear, PFD is considered multimodal, inhibiting both TGF-β synthesis and TGF-β-induced collagen production. Preclinical models of fibrosis across multiple organs have shown that PFD can induce cell-cycle arrest and reduce fibroblast proliferation by down-regulating phosphorylated Smad3, α-SMA, S100A4, fibronectin, and components of the MAPK signalling pathway [157,158,159]. Its anti-inflammatory effects are linked to decreasing production of pro-inflammatory cytokines such as TNFα, IL-1β, and IL-6 [159,160]. Since receiving Food and Drug Administration (FDA)-approval for the treatment of IPF [75,161], PFD is being evaluated for its antifibrotic potential in other organ systems [159], and these properties may also be relevant within the cochlea.

### 7.3. Wnt Pathway

Significant crosstalk between the TGF-β and Wnt pathways amplifies fibrosis by promoting fibroblast activation, myofibroblast differentiation, ECM production, and reduced ECM degradation, creating a self-sustaining feedback loop [162,163,164]. TGF-β upregulates Wnt ligands (e.g., Wnt1, Wnt3a, Wnt5a, Wnt10) and receptors (Frizzled and low-density lipoprotein receptor-related protein (LRP)5/6), stabilises β-catenin through Smad-dependent and independent mechanisms, and suppresses Wnt antagonists such as Dickkopf and Secreted Frizzled-Related Proteins. Wnt signalling, in turn, increases TGF-β receptor expression and promotes TGF-β production, enhancing Smad activation. Both pathways converge on shared transcription factors (T-cell factor/lymphoid enhancer factor for Wnt and Smads for TGF-β) to co-regulate pro-fibrotic genes including ACTA2, COL1A1, FN1 and tissue inhibitor of metalloproteinase 1 [162,163,164].

Evidence for the antifibrotic potential of Wnt inhibition comes from multiple preclinical and early clinical studies. Wnt/β-catenin signalling is consistently elevated in IPF lungs and fibrosis models [165,166,167], and its inhibition reduces myofibroblast differentiation and bleomycin-induced pulmonary fibrosis in mice [168,169]. Wnt10A is overexpressed in IPF, bleomycin models and predicts acute exacerbations [170]. The β-catenin inhibitor PRI-724 decreases fibrosis in bleomycin-treated lungs [171] and reduces hepatic stellate cell activation and ECM deposition in liver fibrosis models [172,173].

Porcupine inhibitors, which block secretion of all Wnt ligands, show broad antifibrotic activity. Wnt-C59 reduces kidney fibrosis in unilateral ureteral obstruction and cardiac fibrosis models [174,175], while GNF6231 mitigates skin fibrosis and pulmonary fibrosis in graft-versus-host disease, with evidence of reversing established fibrosis [176]. Although antibodies such as vantictumab successfully inhibit Wnt signalling via Frizzled receptors in cancer studies, toxicity has limited their use in fibrosis [177].

In the cochlea, Bas et al. identified Wnt signalling as a driver of trauma-induced fibrosis following implantation, contributing to myofibroblast differentiation and ECM production [56]. Targeting Wnt signalling may therefore help reduce cochlear implant-associated fibrosis and improve device performance, although further work is needed to enable clinical translation.

### 7.4. Tyrosine Kinases

TKs are enzymes that function by transferring a phosphate from an ATP molecule to a tyrosine residue on target proteins, activating them via phosphorylation. They are classified as receptor TKs, membrane receptors activated by growth factor binding (e.g., PDGF, FGF, vascular endothelial growth factor [VEGF]), and non-receptor TKs, which function within the cytoplasm [178]. Most fibrosis-related TKs are receptor kinases regulating cell division, metabolism, migration, adhesion, survival, and apoptosis [179,180]. Their central role in disease has driven the development of TK inhibitors (TKI). Because many TKs share structural similarity, single TKIs often inhibit multiple kinases with varying selectivity. Initially developed for cancer, TKIs are increasingly being explored for fibrotic diseases [180].

NTD is a triple TKI targeting VEGF, PDGF and FGF, approved by the FDA for fibrotic lung diseases [161,181]. It also inhibits phosphorylation of Smad3, STAT3, NF-κB, and Src family kinases [182], thereby blocking fibroblast-myofibroblast differentiation and reducing collagen production [183]. NTD exerts anti-inflammatory effects by shifting macrophage phenotype through indirect inhibition of PI3K/AKT and MAPK/ERK signalling [184] and suppresses Wnt3a-induced myofibroblast activation via the Src/β-catenin pathway [185].

As with PFD, interest has grown in NTD’s broader antifibrotic potential. Beyond lung disease, studies are exploring its efficacy in liver, kidney, skin, heart, and eye fibrosis [182,186], with possible relevance to the cochlea.

Imatinib is a small-molecule inhibitor that targets several TKs, including colony-stimulating factor 1 receptor, ABL, c-KIT, FLT3 (CD135), and PDGF receptor-β. It was first recognised for its antifibrotic potential when patients with chronic myeloid leukaemia treated with imatinib showed regression of bone marrow fibrosis [187]. It was later shown to inhibit fibrosis across multiple animal models, prompting clinical trials in human fibrotic diseases, particularly those affecting the lung. However, results from these trials were highly variable, with an improvement in lung function and reduction in skin fibrosis in some patients with SSc [188,189] but no significant impact in IPF [190]. Imatinib can also improve renal function in patients with nephrogenic systemic fibrosis [191].

Dasatinib is a TKI primarily used to treat chronic myelogenous leukaemia and acute lymphoblastic leukaemia. Dasatinib has also shown antifibrotic potential in preclinical studies. It inhibits TGF-β-mediated EMT in lung epithelial cells and reduces lung fibrosis in mouse models by suppressing TGF-β-induced Smad signalling [192]. Dasatinib also reduces fibrosis in other organs such as in uterine fibrosis [193]. This effect is often enhanced when used in combination with quercetin, as both act as senolytics to selectively eliminate senescent cells, which are thought to play a critical role in the development and progression of fibrosis [89]. Despite promising preclinical findings, a recent phase 1 clinical trial evaluating the combination of dasatinib and quercetin in patients with IPF did not show improvements in lung function. However, the study was underpowered, limiting the strength of any clinical conclusions [194].

There are many other TKIs being investigated as antifibrotics including gefitinib, erlotinib, sunitpinib, adavosertib, lapatinib and nilotinib. Many are being used in clinical trials alone or in combination with other treatments. However, the question remains whether these TKIs have therapeutic effects on cochlear fibrosis following cochlear implantation.

## 8. Therapeutic Strategies and Challenges: Implications for the Ear

Treating or preventing fibrosis in the inner ear faces two main challenges. Firstly, effective delivery of pharmacological agents to the inner ear is challenging due to its rigid bony capsule, which isolates the fluids of the membranous labyrinths from the middle ear and surrounding skull, including the brain, thereby limiting drug access. In addition, the inner ear fluids are separated from the systemic circulation by the blood-labyrinth barrier which is largely impermeable to most drug classes [195]. The second challenge is the potential for toxicity to the neural elements of the inner ear [196]. Ototoxicity effecting residual hair cells and functioning spiral ganglion cells is not desirable even in profoundly deafened patients, as the loss of these elements may worsen hearing outcomes and even prevent the potential for future use of regenerative therapies [197]. In cochlear implant recipients with partial deafness, preserving residual low-frequency hair cell function and hearing is essential [198]. Therefore, any treatment with potential ototoxic effects would significantly limit its clinical applicability.

### Pharmacological Delivery to the Inner Ear

Several methods of drug administration could be used to deliver adjuvant therapies at the time of cochlear implantation; however, each method presents unique benefits and challenges. When delivering therapies to inner ear targets, systemic administration, although straightforward, results in relatively low inner ear bioavailability because of the restrictive blood-labyrinth barrier [195]. Therefore, this approach requires higher dosing, which increased risk of off-target effects. Local delivery of pharmacological agents would be preferable, although also not without challenges. Multiple strategies are available for delivering pharmacological agents to the inner ear, including bulk delivery to the middle ear for diffusion across to the round and oval windows, direct application to the RW membrane, and intracochlear delivery via either the RW or a cochleostomy.

Delivery of pharmacological agents into the middle ear generally involves trans-tympanic injections which perforate the tympanic membrane [199,200,201,202]. Although tympanic membrane healing can take several weeks in some individuals, clinical studies indicate that even repeated injections are generally well tolerated [203]. However, the presence of a substance in the middle ear does not guarantee its effective entry into the inner ear. Firstly, the eustachian tube may present an unintentional exit route from the middle ear [201]. Indeed, when intratympanic injections are used on patients with eustachian tube dysfunction, the result is often better as the drug is present in the middle ear for longer [204]. Secondly, agents must penetrate the oval or RW to enter the inner ear, but this can be impeded by false RW membranes, which occur in up to 20% of patients [205]. Moreover, application to the middle ear does not guarantee that the compound will contact these membranes. Achieving stable membrane contact requires either filling the middle ear cavity with large volumes or using novel delivery systems designed to maintain sustained contact with the RW or oval window [206]. Some of these drug delivery systems, such as gels, aerosols, osmotic pumps or sponges, also reduce eustachian tube clearance [126,203,207,208,209,210]. In addition, many studies have investigated the feasibility of direct application to the RW membrane. This can be achieved via intratympanic injection with delivery to the RW niche [202,211], though most studies use a bullostomy to access the RW [209,212,213,214,215]. Direct application of a drug to the RW allows the use of smaller volumes than bulk middle ear delivery, increases the likelihood of inner ear entry, and reduces clearance via the eustachian tube. Studies have confirmed that direct application to the RW can enhance bioavailability in the inner ear [202,216,217] and result in higher levels and faster entry compared to systemic delivery [218,219]. Third, not all pharmacological agents readily cross the RW membrane. Their ability to do so depends on hydrophobicity, molecular size, concentration gradient, and contact duration. Various strategies have been explored to enhance drug transfer across the RW, including encapsulating hydrophilic therapeutics in nanoparticles, using adjuvants or ultrasound-guided microbubbles to increase RW permeability, and employing hydrogels, sponges, Gelfoam^™^ (Pharmacia & Upjohn Co., Kalamazoo, MI, USA), or osmotic mini-pumps to prolong contact duration [157,220,221,222,223]. These methods have successfully increased drug delivery to the inner ear [207,224,225,226,227,228,229], although most drugs have limited bioavailability within the cochlea [227,230,231,232]. Any method applying drugs directly to the RW carries an increased risk of damage to the RW compared with generic middle ear delivery. Finally, the most efficient way to deliver therapeutic agents to the inner ear is direct intracochlear application, either via the RW or a cochleostomy [233]. Cochleostomy ensures precise delivery into the perilymph with controlled concentrations. However, direct application can compromise cochlear integrity due to increased intracochlear pressure, alterations in perilymph ionic composition, and inflammation. In addition, intracochlear fluid dynamics limit dispersion [234], concentrating the pharmacological agent at the site of administration, which is generally near the base of the cochlea. Strategies to optimise delivery throughout the cochlea include apical delivery via a pipette sealed into a fenestra and adjusting the osmotic pump rate using an algorithm [234]. Another approach to address fibrosis around cochlear implants is the use of drug-eluting electrodes. This method provides sustained, targeted delivery over time, allowing intervention in both early and late stages of fibrosis.

## 9. Conclusions

This review highlights numerous therapeutic targets, drawing on extensive research from other organs and tissues. Any candidate must be non-ototoxic and deliverable in a sustained manner without causing mechanical damage to the cochlea. Additionally, accurate quantification of fibrosis is essential to assess effect size relative to the only clinically used drug, corticosteroids, alongside evaluation of cochlear function outcomes.

## Figures and Tables

**Figure 1 cells-14-01924-f001:**
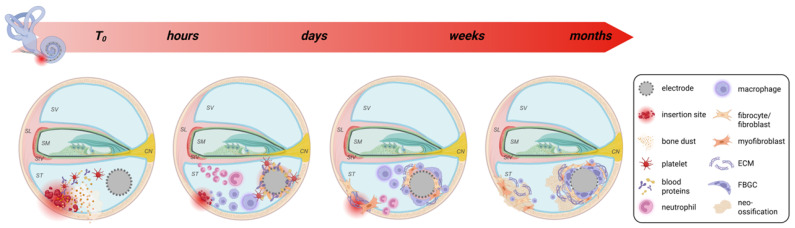
The development of fibrosis and neo-ossification after cochlear implantation. The response to cochlear implantation begins at the time of implant insertion (T_0_), where surgical trauma and cochlear implantation trigger an acute inflammatory response. Blood proteins and platelets accumulate at the insertion site and also adsorb to the surface of the implant forming a provisional matrix. Phagocytic cells including neutrophils and macrophages migrate to the site of injury to clear any cellular debris and adhere to the provisional matrix surrounding the electrode tract where they produce and secrete cytokines which promote wound healing and the foreign body response (FBR). Specifically, these cytokines trigger the activation of macrophages, which fuse to form foreign body giant cells (FBGCs) as well as the migration, proliferation and differentiation of fibroblasts/fibrocytes, cells which are responsible for the deposition of extracellular matrix (ECM) leading to the fibrous encapsulation and subsequent neo-ossification around the electrode. SV: scala vestibuli, SM: scala media, ST: scala tympani, SL: spiral ligament, StV: stria vascularis, CN: cochlear nerve. Created with Biorender.

**Figure 2 cells-14-01924-f002:**
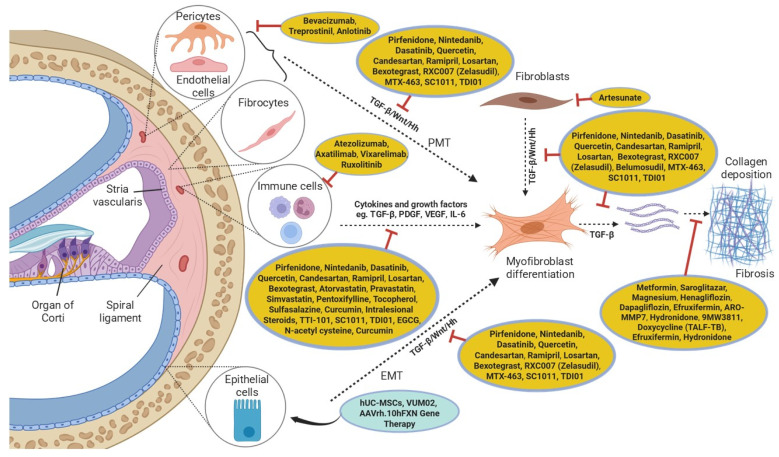
Pathogenic pathways of cochlear fibrosis and drugs in various stages of development or clinical use, focusing on different pathways that converge in fibrotic diseases such as pulmonary and cardiac fibrosis, liver cirrhosis, and systemic sclerosis. Each drug targets specific biological mechanisms contributing to the fibrotic process. Hh: Hedgehog, MSCs: mesenchymal stem cells, PMT: Pericyte to mesenchymal transition. Created with Biorender.

## Data Availability

No new data were created or analyzed in this study.

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
