# Peer review of "Understanding the Mechanisms Driving Fibrosis Following Cochlear Implantation—Lessons from Other Tissues"

_cells, 2025, doi:10.3390/cells14231924_

Round 1
Reviewer 1 Report
Comments and Suggestions for Authors
Understanding the mechanisms driving fibrosis following
Cochlear Implantation - Lessons from Other Tissues is a review of causes and strategies to avoid inner ear fibrosis or ossification in cochlear implantation surgery.
The authors thoroughly described the topic with a consistent number of recent references.
The central limitation of the paper is its readability, as it is too long and reports sometimes unnecessary details. It is recommended to have a drastic reduction, particularly in the chapter about new therapies, because this form is not inviting for interdisciplinary reading, especially for ENT and audiologists. To improve clarity, we suggest segmenting the chapters into subheadings.
Another point to modify, specifically by shortening, is the description of general fibrosis, as it is redundant, while the cochlear reference is only occasional.
Moving on to the contents, it is important to outline the difference between the "natural" fibrosis creating a capsule to wrap the electrode because it is a foreign body versus a massive (pathological) fibrosis even with ossification that could be related to two causes: excessive surgical trauma, intracochlear bleeding (iatrogenic cause) and/or excessive reaction to the foreign body (patient reaction). Please check the following papers: doi: 10.1002/lio2.329; doi: 10.1016/j.heares.2022.108681; doi: 10.1016/j.heares.2022.108536; doi.org/10.1371/journal.pone.0136617.
Therapeutic agents should aim to limit excessive fibrosis, as the capsule is physiological and can create problems only in the event of reintervention.
We recommend a brief introduction to the topic, limited to the clinical effects of fibrosis and ossification on outcomes.
The fibrosis should be treated in consideration of the topic, distinguishing the iatrogenic effects of surgery (a) and the presence of a foreign body (b). The other causes of cochlear fibrosis are off topic. About the ossification, there is no mention of the presence of free blood inside the inner ear https://doi.org/10.1371/journal.pone.0136617.
It is not easy to understand the utility of the chapter about fibrosis in other tissues, and we suggest canceling it.
Another point to review is the chapter on mechanisms of fibrosis, as a list of pathologies is presented without relation to the topic. Additionally, the authors did not describe the reaction to a foreign body as a pacemaker, considering that in both cases, the body responds with a fibrotic capsule. Consulting cardiologic literature could be of some interest.
In the chapter on current prevention and treatments, the corticosteroid treatment is described. However, there is no distinction between local and general administration, the time in relation to the surgery, and the posology.
About novel therapeutic approaches, we already gave a suggestion, but it is better to reply that this chapter is the heavier part of the paper. We suggest reducing the length by two-thirds, considering that the reference list and Figure 2 are already exhaustive and complex for those who prefer to delve deeper into the topic.
Finally, in paragraph 8.1, the authors list a series of possible sites to administer drugs to treat inner ear pathologies, but could these solutions be proposed in the presence of a cochlear implant? Important to outline that there is no mention about the methodology of administering intracochlear corticosteroids, for example, during surgery.
In conclusion, the paper needs major revisions.
Author Response
Please find uploaded a revised manuscript of the review article “Understanding the mechanisms driving fibrosis following cochlear implantation - lessons from other tissues” with Cecilia Prêle as the corresponding author, which we would like to have considered for publication in Cells.
This invited review explores the concept of implant-induced cochlear fibrosis, examining the underlying mechanisms and drawing on insights from fibrosis in other organs to accelerate the development of novel therapeutic strategies aimed at improving cochlear implant outcomes.
All authors have read the manuscript and reviewer’s comments and have approved its resubmission.
We would like to thank the editor and reviewers for their comments which have significantly improved this manuscript.
More specifically:
- We have revised the abstract to clarify the aim of the study, which is to summarise current knowledge from studies of fibrosis in other organs and apply these insights to the development of novel therapeutic strategies for cochlear fibrosis.
Reviewer 1 comments and suggestions for authors
- The central limitation of the paper is its readability, as it is too long and reports sometimes unnecessary details. It is recommended to have a drastic reduction, particularly in the chapter about new therapies, because this form is not inviting for interdisciplinary reading, especially for ENT and audiologists. To improve clarity, we suggest segmenting the chapters into subheadings.
Another point to modify, specifically by shortening, is the description of general fibrosis, as it is redundant, while the cochlear reference is only occasional.
We carefully considered the reviewer’s suggestion to remove sections discussing general fibrosis, non-cochlear fibrosis, and drug targets. However, the central purpose of this review is to synthesise existing knowledge on the development and progression of fibrosis in other organ systems and to illustrate how these insights can inform our understanding of fibrotic processes within the inner ear. Identifying shared cellular and molecular pathways across different tissues provides an opportunity to accelerate discovery, enabling the more rapid identification, development, and implementation of novel therapeutic strategies to prevent or treat cochlear fibrosis. Therefore, reducing the information described in the manuscript would compromise the overall objective of the review, which is to provide a comprehensive, cross-disciplinary framework that integrates lessons from other fibrotic diseases to advance understanding and guide future research directions in cochlear fibrosis.
- Moving on to the contents, it is important to outline the difference between the "natural" fibrosis creating a capsule to wrap the electrode because it is a foreign body versus a massive (pathological) fibrosis even with ossification that could be related to two causes: excessive surgical trauma, intracochlear bleeding (iatrogenic cause) and/or excessive reaction to the foreign body (patient reaction). Please check the following papers: doi: 10.1002/lio2.329; doi: 10.1016/j.heares.2022.108681; doi: 10.1016/j.heares.2022.108536; doi.org/10.1371/journal.pone.0136617.
We agree with the reviewer that following cochlear implantation fibrosis occurs as a result of both the trauma induced in response to injury (iatrogenic) and as part of the FBR to the implant itself. This is stated in paragraph 2 page 3. Post mortem temporal bone studies have revealed that almost all individuals who have undergone cochlear implant surgery have some degree of fibrosis. Furthermore, they demonstrate that this fibrosis can be extensive almost completely filling the scalae in some individuals, a process which far exceeds encapsulation alone and which may indicate a progressive fibrotic response. The review article by Foggia et al and the recent paper by Geerarydn et al, are already cited. The paper by Ryu et al has been added (section 6.2 para 1). Rahman et al 2022 is a review paper and we have opted to prioritise original articles.
- Therapeutic agents should aim to limit excessive fibrosis, as the capsule is physiological and can create problems only in the event of reintervention.
Several studies linking the negative effects experienced post cochlear implantation to fibrosis and neo-ossification are summarised on page 4. These clearly demonstrate that the effects of fibrosis may extend beyond complicating revision surgery but may have a significant impact on residual hearing loss and implant function. We acknowledge that extent to which fibrosis impacts cochlear implant function remains unclear.
- We recommend a brief introduction to the topic, limited to the clinical effects of fibrosis and ossification on outcomes.
The effect of fibrosis on clinical outcome, based on the limited studies available is covered on Page 4.
- The fibrosis should be treated in consideration of the topic, distinguishing the iatrogenic effects of surgery (a) and the presence of a foreign body (b). The other causes of cochlear fibrosis are off topic. About the ossification, there is no mention of the presence of free blood inside the inner ear https://doi.org/10.1371/journal.pone.0136617
The paper by Ryu et al has been added and discussed on page 8, paragraph 4, ref 113.
- It is not easy to understand the utility of the chapter about fibrosis in other tissues, and we suggest cancelling it.
The purpose of this review was to gain insight from mechanistic studies performed in other tissues. Given the limited information available in the field of cochlear implant fibrosis, we believe this adds significant value to this manuscript.
- Another point to review is the chapter on mechanisms of fibrosis, as a list of pathologies is presented without relation to the topic. Additionally, the authors did not describe the reaction to a foreign body as a pacemaker, considering that in both cases, the body responds with a fibrotic capsule. Consulting cardiologic literature could be of some interest.
A section detailing similarities in fibrotic encapsulation of cardiac pacemaker has been added on page 6, last paragraph.
- In the chapter on current prevention and treatments, the corticosteroid treatment is described. However, there is no distinction between local and general administration, the time in relation to the surgery, and the posology.
The method of corticosteroid delivery has been described please see section 6.2. We have further clarified the mode of delivery when needed. We believe that discussing posology is beyond the scope of the review.
- About novel therapeutic approaches, we already gave a suggestion, but it is better to reply that this chapter is the heavier part of the paper. We suggest reducing the length by two-thirds, considering that the reference list and Figure 2 are already exhaustive and complex for those who prefer to delve deeper into the topic.
We thank the reviewer for their suggestions but feel that understanding the mechanisms of action of potential drug targets is important when considering suitable candidates for the inner ear.
- Finally, in paragraph 8.1, the authors list a series of possible sites to administer drugs to treat inner ear pathologies, but could these solutions be proposed in the presence of a cochlear implant? Important to outline that there is no mention about the methodology of administering intracochlear corticosteroids, for example, during surgery.
We have added a sentence to provide additional context, highlighting the utility of drug delivery methods at the time of cochlear implant surgery (Page 14, Section 8.1).
The method of corticosteroid delivery is described in section 6.2.
Reviewer 2 Report
Comments and Suggestions for Authors
This is a review paper on etiopathogenesis of cochlear fibrosis after cochlear implantation. The title and abstract are fine, and the title needs to reflect type of study .
In the first sentence, one can hardly say that performance is highly variable and largely unexplained, since we already know about the majority of factors influencing the results, but just can't do much about them (yet); candidate age, electrode design and length, and electroacoustic stimulation limitations. Tissue fibrosis is important, but certainly not the most important variable.
The rest of the introduction is fine.
The section on fibrosis introduces contrary evidence on models that have little to do with CI, and is best moved to the discussion section or shortened.
This review is either a narrative review, or a scoping review, and should follow PRISMA gudidelines in either case, with a literature search methodology section and PRISMA flowchart.
The discussion on intratympanic and intracochlear delivery is succint and well organized. Recent literature has been cited.
I would suggest a minor revision primarily addressed at argument flow and clarity of presentation. PRISMA methodology would go a long way to establish which studies have been cited, since many show opposing results, especially regarding outcomes, where impendance is not a clinical outcome, but a measure of fibrosis not directly related to the true outcome that patients and clinicians care about most - speech recognition. I would suggest adding another section trying to establish fibrosis and ossification as a factor in WRS outcomes to make the paper relevant to a wider audience.
Author Response
Please find uploaded a revised manuscript of the review article “Understanding the mechanisms driving fibrosis following cochlear implantation - lessons from other tissues” with Cecilia Prêle as the corresponding author, which we would like to have considered for publication in Cells.
This invited review explores the concept of implant-induced cochlear fibrosis, examining the underlying mechanisms and drawing on insights from fibrosis in other organs to accelerate the development of novel therapeutic strategies aimed at improving cochlear implant outcomes.
All authors have read the manuscript and reviewer’s comments and have approved its resubmission.
We would like to thank the editor and reviewers for their comments which have significantly improved this manuscript.
More specifically:
- We have revised the abstract to clarify the aim of the study, which is to summarise current knowledge from studies of fibrosis in other organs and apply these insights to the development of novel therapeutic strategies for cochlear fibrosis.
Reviewer 2 comments and suggestions for authors
- This is a review paper on etiopathogenesis of cochlear fibrosis after cochlear implantation. The title and abstract are fine, and the title needs to reflect type of study.
This review is a narrative review which summaries the current literature.
- In the first sentence, one can hardly say that performance is highly variable and largely unexplained, since we already know about the majority of factors influencing the results, but just can't do much about them (yet); candidate age, electrode design and length, and electroacoustic stimulation limitations. Tissue fibrosis is important, but certainly not the most important variable.
We have updated the first sentence of the Introduction to reflect the reviewers suggestion. This change is also highlighted below.
Hearing restoration surgery has advanced significantly since the development and implantation of the first cochlear implant in 1961 [1], but variability in speech perception outcomes has been reported, with the causes of this variability not yet fully understood [2].
- The section on fibrosis introduces contrary evidence on models that have little to do with CI, and is best moved to the discussion section or shortened.
It is not clear exactly what the reviewer is referring to. In most part, the section refers to commonalities between mechanisms of fibrosis between organs.
- This review is either a narrative review, or a scoping review, and should follow PRISMA guidelines in either case, with a literature search methodology section and PRISMA flowchart.
PRISMA guidelines are generally used for meta-analyses and systematic reviews. As this is a narrative review we have not included a methodology section.
- I would suggest a minor revision primarily addressed at argument flow and clarity of presentation. PRISMA methodology would go a long way to establish which studies have been cited, since many show opposing results, especially regarding outcomes, where impendance is not a clinical outcome, but a measure of fibrosis not directly related to the true outcome that patients and clinicians care about most - speech recognition. I would suggest adding another section trying to establish fibrosis and ossification as a factor in WRS outcomes to make the paper relevant to a wider audience.
Fibrosis and it impact on hearing outcomes is discussed on Page 4. The primary aim of this review is to focus on the molecular and cellular drivers of fibrosis to identify drug candidates for prevention of fibrosis in the inner ear. Therefore, we have limited the clinical impact section. It should also be noted that currently, studies identifying audiological markers of fibrosis and the studies demonstrating the direct clinical impact of fibrosis remain limited.
Round 2
Reviewer 1 Report
Comments and Suggestions for Authors
I have reviewed the manuscript, “Understanding the mechanisms driving fibrosis following cochlear implantation – lessons from other tissues,” following the authors’ recent modifications.
Unfortunately, the suggestions to reduce the text were not accepted. In light of this, I recommend rejection of the paper. The current length and level of detail are more suitable for an academic book than for a scientific journal, which is intended for updates rather than comprehensive explanations of basic science.
Author Response
In response to both reviewer 1 and the editor's comments we have made some significant changes to the manuscript. We have not deleted sections related to 'other tissues' as the point of this review is to take what is learnt from other tissues to help identify possible therapeutic approaches to treat implant-induced cochlear fibrosis. However, we have significantly reduced the length of the manuscript, with most of the editing in the sections related to fibrosis in other tissues and therapeutic approaches. We have not changed what we said, just cut some of the detail. The manuscript has been shortened in length by approximately 20%, from about 7820 to 6195 words (not including abstract or refs).